# Benefits, Risks and Gender Differences in Sport, and Exercise Dependence: Key Role of Alexithymia

**DOI:** 10.3390/ijerph19095288

**Published:** 2022-04-26

**Authors:** Paola Manfredi

**Affiliations:** Department of Clinical and Experimental Sciences, University of Brescia, 25123 Brescia, Italy; paola.manfredi@unibs.it

**Keywords:** sport, exercise dependence, alexithymia, gender differences, young adults, well-being, exercise addiction

## Abstract

The literature has long highlighted the benefits of sport, but too much sport could indicate a dependence without a substance, namely exercise dependence. The literature has only recently investigated this issue and therefore several questions are open, particularly with regard to psychopathological significance and gender differences. The aim of this paper is to investigate whether young subjects (M = 20 years) with a risk of exercise dependency and non-dependent symptomatic display other behavioural and psychological suffering and discomforts, or whether such an investment in physical activity is compatible with a framework of relative well-being; and if there are differences related to gender. A total of 396 subjects were involved in this study. Exercise Dependence Scale, Toronto Alexithymia Scale, Subjective Happiness Scale, Satisfaction with Life Scale and an ad hoc questionnaire with information relating to the consumption of alcohol, soft and hard drugs, quality of sleep and nutrition, affective and friendship relationships, hobbies, presence of psychological or physical disorders, motivations for sporting activity, and any traumatic experiences were used. With regard to the Exercise Dependence Scale (EDS), the prevalence of subjects at risk of exercise dependence was 1.5% and that of non-dependents symptomatic was 31.3%. Considering only men, the percentage of subjects at risk of exercise dependence rose to 3% and that of non-dependents symptomatic to 47.9%. Our data support the hypothesis that the risk of exercise dependence and the symptomatic condition without dependence can occur in situations of relative well-being (happiness, satisfaction, relationships) without significant associations with other problematic behaviours. Modest consumption of soft drugs is reported in non-dependent symptomatic subjects. The hypothesis of increased levels of alexithymia is confirmed but limited to the male gender. Gender differences are confirmed in the frequency and motivation to practice sport and in the risk of dependence. It is important that the assessment of addiction risk is integrated with the assessment of alexithymia and personal and social resources over time.

## 1. Introduction

Sporting activity has been in our culture since ancient Greek civilization and is linked to a concept of health and well-being. In recent decades, the scientific literature has highlighted the association between exercise and physical and mental health [1] quality of life [2], stress [3] mood [4,5,6], anxiety and depression [5,6,7,8], and recent studies have shown that physical activity activates reward circuits [9] and increases peripheral levels of dopamine [10], noradrenaline, and serotonin and endorphin release [11,12].

In recent decades, it also emerged that there can be an excess of this activity, which can then take on the characteristics of a dependence or an addiction without a substance [13,14]. The term of exercise dependence was first used in 1969 [15], but to this day there is some confusion about the terms and definitions referring to this behaviour. For example, there is talk of “excessive exercise” [16]; “obligatory exercise” [17]; “exercise abuse” [18]; “exercise dependence” [13,19]. “exercise addiction” [20]. However, the prevailing terms are dependency and addiction. Allegre et al. [21] defined two different conceptualizations of primary dependence based on diagnostic criteria or behavioral components of dependence. Exercise dependence is “a craving for leisure time physical activity that results in uncontrollable excessive exercise behavior and that manifests in physiological and/or psychological symptoms” [13]. The diagnostic criteria of exercise dependence are based on the diagnostic criteria for substance dependence of DSM-IV. These are tolerance, withdrawal effects, intention effect, lack of control, time, reductions in other activity, and continuance. The Exercise Dependence Scale is based on these criteria. 

Addictive behavior is defined as a repetitive pattern, experienced as loss of control, that increases the risk of disease and personal and social problems. The components of exercise addiction are salience, mood modification, tolerance, withdrawal symptoms, conflict, and relapse. Exercise Addiction Inventory is based on these six components. 

Allegre et al. [21] identified three common elements between the two conceptualizations: neuroadaptation, salience and adverse consequences. They hypothesized that these three common features explained the same prevalence of exercise addiction in people (less than 5%). A further complication is that the terms addiction and dependence can often be used interchangeably [21] and in many works is used EDS but the subjects are defined as addicted, e.g., [22,23,24], or as having compulsive exercise [25].

Each form of dependence or addiction places constraints on the subject and thus limits potentially healthier expressions. The activity or object of dependence becomes central and this compromises lifestyles, social relationships, family life, emotional attachments and often, even, a reduction in health status and occupational impoverishment [15,16].

The DSM-5 recognised the existence of behavioural, substance-free addictions, but only considered gambling, and therefore does not explicitly include exercise addiction/dependence. However, in a systematic review in which the most frequently used instrument to assess the problematic behaviour was the Exercise Dependence Scale [26], associations with anxiety, depression and eating problems psychological disorders were found; exercise dependence was also found to be associated to compulsive buying, bulimia, and, in a lesser extent, hypochondria [27,28], psychiatric disorder [22,23] depressive disorders, personality disorders and obsessive-compulsive disorders [22], loneliness and anxiety [24], and cognitive and behavioural inflexibility [29].

It is important to make some distinctions with respect to forms of addiction, dependence, age, and gender. There are forms, such as internet gaming behaviour symptom networks that mixed significantly with other addictions [30]. Some forms of addiction, such as drug and alcohol addiction [31] and gambling addiction [32,33] may have adolescent beginnings and become entrenched in later ages, but it is believed that exercise addiction/dependence deserves specific consideration, especially when it involves late adolescents/young adults. Several contributions in the literature emphasised the need to understand forms of addiction/problematic behaviours as specific disorders, rather than an overarching addiction diagnosis [30,34]; the recognition of different psychological dynamics beyond symptoms is also necessary for a targeted clinical approach [35].

With respect to age, in the (late) adolescent period it is not easy to distinguish indicators of psychopathology that will remain, from transitory symptoms, linked to the specific developmental crisis, for which a diagnostic label is not appropriate. In the adolescent period—compared to other age groups—there is a greater investment in sporting activity; possible exercise dependence could therefore represent a sort of buffer against problems and developmental crises, which could in time normally resolve themselves, together with a reduction in dependence or excessive sporting activity. Amongst other things, there are contributions in the literature that interpreted the addictive disorders (e.g., substance use disorders) as an attempt at self-care by the subject [36], and dependences as analgesia against pain [37]. A fortiori it seems to us that this could be applied to exercise dependence and to developmental phases, although an ideal dimension of health and well-being does not contemplate either dependence or addiction.

Within this perspective, the investigation of gender differences is particularly important. In the field of behavioural dependence, the exploration of gender differences is still under-researched, although the obvious differences between the sexes, in psychological, cultural, social, biological, endocrine terms [38], justify the need for studies in this area.

Given that (a) in a still evolutionary phase some symptomatic behaviours, among which addictions, can be transitory and not indicative of structural psychopathologies, but of healable fragility (b) sport is considered a healthy activity (c) among the forms of dependence, exercise dependence seems to be the one with the least negative impact on health, our hypothesis is that in the sample of university students there are advantages in those who practise sport compared to those who do not, and that the condition of non-dependence symptomatic and risk for dependence does not correlate with other indicators of discomfort and does not differentiate the sample with respect to general conditions of well-being. 

Considering also that addiction can be a treatment response with respect to a subject’s fragility, we hypothesise that it is more accessible to subjects who are less able to regulate their emotions, therefore we hypothesise that non-dependent symptomatic conditions and risk for addiction correlate with an alexithymic tendency. 

Given that at a social level sporting activity is still more encouraged in men than in women, we expect gender differences in both frequency and motivation. 

We therefore hypothesise that (A) we do not expect to find comorbidity with other problematic behaviours, that could become dependence, such as alcohol, smoking, drugs, and eating disorders; (B) we do not expect significant differences with respect to happiness and satisfaction, presence of intimate and friendship relationships, presence of other hobbies or recreational activities, in risk for exercise and non-dependence symptomatic subjects; (C) we should find in symptomatic and at risk subjects higher values of alexithymia; (D) we expect to find differences between males and females and in particular a greater presence of men both in sport and in the risk of exercise addiction and differences in motivation to practice sport; ad (E) we should find in both sexes some advantages (happiness and satisfaction) related to sport compared to those who do not practise sport.

## 2. Materials and Methods

The study was conducted in accordance with the Declaration of Helsinki and approved by the university committee. The project was presented, in pre-COVID period, to students of the three-year degree courses in exercise science and nursing. The questionnaires were anonymous, and participation was voluntary. Informed written consent was obtained from all subjects involved in the study. Inclusion criterion was age between 18 and 25 years. The exclusion criterion was age over 25 years. The choice of this age group was motivated by the idea of identifying subjects, where the choice of sport was as attributable as possible to the subjects themselves. At a younger age there may be pressure from, for example, the family, and sporting activity is often more accessible and supported by educational or municipal institutions. At the age of twenty, sport is usually a more conscious and deliberate choice. A total of 420 protocols were collected and 24 protocols were subsequently excluded since they were filled in by older students.

For the assessment of exercise addiction Exercise Dependence Scale (EDS) [13,39] was used. Downs et al. [40] indicated the following parameters of the EDS: Tucker-Lewis Index = 0.96, comparative fit index = 0.97, root mean square error of approximation = 0.05, mean absolute standardised residual = 0.02. 

The scale consists of 21 items, for each of which a score is assigned based on a six-point Likert scale, where “1” corresponds to never and “6” to always. The test is divided into seven scales according to the criteria for defining addiction [13]. They are: (1) tolerance: either a need to increase the amount of exercise to achieve the desired effect or a diminished effect occurs with continued use of the same amount of exercise; (2) abstinence: manifested either by withdrawal symptoms characteristic for exercise (e.g., anxiety, fatigue) or the same amount of exercise (or a closely related amount) is taken to relieve or avoid withdrawal symptoms; (3) intention effect: exercise is often taken in larger amounts or for a longer period than intended; (4) lack of control: A persistent desire or unsuccessful effort to reduce or control exercise; (5) time: a large amount of time is spent on activities necessary to achieve exercise (e.g., exercise holidays); (6) reduction of other activities: social, occupational or recreational activities are abandoned or reduced due to exercise; (7) continuity: exercise is continued despite the knowledge that one has a persistent or recurrent physical or psychological problem that is likely to have been caused or exacerbated by exercise (e.g., continuing to run despite an injury”) [13,39]. The test differentiates between “at risk for exercise dependence”—with a score of 5 to 6 in at least three of the seven criteria—“non-dependent symptomatic”—with a score of 3 to 4 in at least three criteria—and “non-dependent asymptomatic”. The Exercise Dependence Scale, although a validated test, is not a tool for clinical diagnosis, but provides an adequate indication for research purposes.

Alexithymia was assessed with the Toronto Alexithymia Scale (TAS 20) [41]. “(…) the 20-item Toronto Alexithymia Scale (TAS-20) has good internal consistency and test-retest reliability, and a factor structure that reflects three separate, yet conceptually related, facets of the alexithymia construct.” [42] (p. 552). It is a self-report scale consisting of 20 items that are grouped around three factors: (TAS F1) difficulty in identifying and distinguishing feelings and physical sensations; (TAS F2) difficulty in describing feelings; (TAS F3) externally oriented thinking. Although alexithymia in itself indicates a lack of words for emotions, it takes on the meaning of an inadequacy in the regulation of emotions. The questionnaire, whose Italian translation was edited by Bressi et al. [42], is based on a five-point Likert scale. Subjects obtaining scores below 51 are considered non-alexithymics between 51 to 60 subjects are considered borderline, whereas those obtaining scores over or equal to 61 are considered alexithymics subjects.

The Subjective Happiness Scale (SHS) was used to measure the overall level of perceived happiness [43]. SHS is a self-assessment measure consisting of four elements on a 7-step scale (from 1 = I’m not a very happy person to 7 = I’m a very happy person). Two items evaluate the general perception of happiness and the other two evaluate the perception of happiness compared to the others. The total score is obtained by adding the scores of the individual items, reaching an overall score that can range from 4 to 28. Higher scores represent greater overall happiness. Confirmatory Factor Analysis of Italian version [44] supported the one-dimensionality of the SHS, with acceptable fit indexes (NNFI = 0.96; CFI = 0.99; RMSEA = 0.08; 95% C.I. [0.04–0.12]). Multi-group analyses supported total invariance of the SHS measurement model for males and females.

The Satisfaction with Life Scale (SWLS) was used to measure satisfaction on one’s life. SWLS [45] is a self-assessment scale that measures general life satisfaction. It is asked to answer the items by filling in a 7-point Likert scale (1 = strongly disagree, 7 = completely agree). Two articles ask respondents to report what extent they consider themselves to be a happy or unhappy person, in absolute terms and relative to other people. The other two elements require indicating the extent to which the two descriptions of happy and unhappy people apply to them. The total scores on the scale are calculated by averaging the responses to the 5 items and therefore range from 1 to 7. A higher score indicates greater overall satisfaction with life. The SWLS is a unidimensional measure; the first factor accounted for 74% of the variance. The mean coefficient alpha was 0.83 [46]. 

In addition, a questionnaire was designed to collect socio-anamnestic data, information on the consumption of alcohol, soft and hard drugs, quality of sleep, eating behaviour, emotional and friendship relationships, hobbies, the presence of mental or physical disorders, traumatic experiences, and motivations to play sport. In particular, the proposed motivations for sport were to keep healthy, fit, increasing muscle mass, meeting people, controlling weight, habit, relieving tension, occupying time, pleasure, and fun, other (please specify). The question about alcohol consumption provided the following options: No, I am a teetotaler or only drink on special occasions; Not now, but I used to drink in the past; Yes, but I usually only drink at meals; Yes, I usually drink outside meals, but in moderation; Yes, occasionally I drink too much but without getting drunk; Yes, I occasionally drink too much and get drunk; Yes, I often drink too much and get drunk.

Smoking habit included the following answers: I don’t smoke or smoke a little; I used to smoke and have recently stopped; I smoke quite a lot The questions on the use of soft and hard drugs included the following response options: No, never; Currently not, but in the past I have used them frequently; Yes, I have tried them a few times; Yes, occasionally; Yes, quite often; Yes, usually. The information about your eating habits was as follows: I often eat outside of meals; I eat quickly and always finish before others do; I only take short breaks during the working day to eat; I have repeatedly suggested eating less in order to reduce my weight; I eat constantly when I am tired or nervous; I eat very little when I am tired or nervous; I usually eat breakfast and main meals; I eat too much, always have a big appetite; I eat very little, never have an appetite; I have tried to follow or adhere to a diet without succeeding. With respect to sleep, the options were: regardless of the hours of sleep, I do not wake up in the morning rested; I usually sleep well and wake up rested. The questionnaire also included a question about experiences which the subject felt were particularly negative or traumatic. If so, the subject was asked to describe them and indicate when they had occurred. It was also investigated whether the subject had a stable emotional relationship and whether or not he or she had had relationships in the past, short and insignificant relationships, significant relationships. With regard to friendships, the questionnaire provided the following response options: I do not have good friends and I am often alone; I have some superficial friendships; I have many friendships but superficial; I have few good friends; I have some deeper and some shallow friendships; I have many good friends. The questionnaire also investigated whether the subject had hobbies, practiced sports, and suffered from mental or physical disorders.

### Statistics Analysis

The database was formatted through the Microsoft-Excel^®^ software (Microsoft Corporation, Redmond, WA, USA) and later imported from the IBM-SPSS^®^ software ver. 26 (IBM SPSS Inc., Chicago, IL, USA).

Normality of the distributions was assessed using the Kolmogorov-Smirnov test. Categorical variables were presented as frequencies or percentages and compared with the use of the Chi-Square test; associations of the crosstabs were verified using standardized adjusted residuals. As the data are on ordinal scales and there is no distributional adequacy (Kolmogorov-Smirnov test), descriptive indices were used, comparisons between groups were made with the use of the Mann-Whitney test and Kruskal-Wallis test, and correlations among variables were determined by the Pearson’s or Spearman’s rank correlation test. A two-sided α level of 0.05 was used for all tests.

## 3. Results

The age of the sample was between 18 and 25 years; μo and Mdn are 20, M = 20.68, and SD = 1.506 (Figure 1).

The sample consisted of 396 subjects of which 42.7% were men and 57.3% women. With respect to the degree course, the distribution was as follows: 51.4% were from the exercise science course and 48.4% from the nursing course. The sample was therefore equally divided between the two courses, but the adherence was higher for the exercise science course, in which 65% of those enrolled responded compared to 42% for the nursing course. In addition to interest, logistical issues played a role in the difference in participation. The sample of those who play sports consisted of 284 subjects, 54.2% men, and 45.8% women.

In our sample the prevalence of subjects at risk for exercise dependence was 1.5% and of non-dependence symptomatic 31.6% (Table 1). Considering only men, the percentage of subjects at risk for exercise dependence rose to 3% and that of non-dependent symptomatic to 47.9%. Considering only playing sport students, symptomatic subjects rose to 42.4% and subjects at risk to 2.1%, while among exercise science students they were 44% and 3.1% respectively.

### 3.1. Comparison between Subjects Who Do and Not Do Play Sport Regularly

Comparing subjects who playing sport with those who do not, there were no significant differences in smoking, alcohol or drug consumption and traumatic experiences; there were no differences in diet, friendships, and emotional relationships.

On the other hand, there were significant differences in alexithymia, in TAS F1 difficulty in identifying and distinguishing feelings and physical sensations (*p* = 0.001), TAS F3 externally oriented thinking (*p* = 0.001), somatic symptoms (*p* = 0.037) SHS Subjective Happiness Scale (*p* = 0.028) and SWLS Satisfaction with Life Scale (*p* = 0.023) and sleep quality (*p* = 0.002). Those who do not play sport had higher values in TAS F1 and lower values in factor TAS F3 (Table 2).

They had more somatic symptoms than those who play sport (23 vs. 21). Moreover, those who play sport had better rest than those who do not (52.3 vs. 35.2).

Those who play sport had, compared to the SHS, higher levels of happiness in (*p* = 0.015) (4.75 vs. 4.5) and greater satisfaction with their lives (*p* = 0.033) (25 vs. 23).

### 3.2. Exercise Dependence Scale

Considering the three categories of the EDS, some differences emerged with respect to TAS F3 (*p* = 0.002) (Table 3).

Statistically significant was the difference regarding somatic symptoms, for which the subjects at risk for exercise dependence had lower scores than the symptomatic non dependence (17 vs. 22). There was also a statistically significant difference (*p* = 0.021) with respect to the use of soft drugs, the use was found more in symptomatic non-dependence subjects (Table 4).

Comparison with the EDS categories showed no differences for smoking, sleep, traumatic experiences, eating behavior, friendships, hobbies (it should also be noted that only one subject that used the internet for 10 h/day), somatic symptoms, levels of satisfaction and happiness.

Traumatic experiences were not related to a dependence. Half of the subjects at risk for dependence declared that they had suffered an injury, but only one specified it (trauma to the jaw and end of competitive activity); non-dependent symptomatic indicated potentially involving traumas (suicide of the neighbor, domestic violence, loss of the best friend), but as many did not specify the trauma it was not possible to make statistical inferences. 

About motivation, the desire to stay healthy and fit (*p* = 0.000), the acquisition of muscle mass (*p* = 0.000), the desire to meet people (*p =* 0.000), and habit (*p* = 0.000) were most frequent in non-dependent symptomatic; the desire to relieve tension (*p* = 0.000) (Table 5) and pleasure (*p =* 0.000) (Table 6) were most frequent in non-dependent symptomatic and at risk for exercise dependence.

### 3.3. Gender Differences

Concerning gender, the following significant differences were noted: alcohol consumption (*p* = 0.000), soft drug consumption (*p* = 0.000), hard drugs (*p* = 0.043), TAS F1 (*p* = 0.000), TAS F2 (*p* = 0.048), TAS F3 (*p* = 0.000), EDS and all its component scales (*p* = 0.000). Women were largely abstemious or drink only on special occasions (57.3%), were asymptomatic with respect to exercise dependence (80.9%), had never used soft drugs (86%) or hard drugs (99.6%). They had slightly higher scores of TAS F1 (16, 28 vs. 13.92) and TAS F2 (13.92 vs. 12.9) t while they had lower scores in TAS F3 (16.58 vs. 18.96).

Considering separately the women’s and men’s samples with respect to the exercise dependence, the women’s sample (also comparing the levels 1 and 2 of the EDS) absolutely no significant differences emerged, with the exception of the past relationships in which the symptomatic persons (*p* = 0.000), declared to have had significant relationships in the past and no short and insignificant relationships, in contrast to the asymptomatic girls who divided themselves quite equally with respect to this theme. Differentiating with respect to sport, it emerged that in those who do not play sport the values of the TAS F1 were higher (M = 15.38 vs. M = 17, 46 *p* = 0.000).

In the men’s sample, the distribution in the exercise categories (EDS) was different for: somatic symptoms (*p* = 0.023), TAS 20 (*p* = 0.000), TAS F1 (*p* = 0.007), TAS F2 (*p* = 0.012), TAS F3 *(p* = 0.029) (Table 7), soft drugs (*p* = 0.028).

While regarding alexithymia there was a progressive increase of the values as the EDS categories progress, about somatic symptoms it was the non-dependent symptomatic who suffered the most, just as regarding soft drugs the non-dependent symptomatic used them more than the subjects at risk and the non-dependent asymptomatic. Differentiating with respect to sport, TAS 20 and TAS F2 values were higher in those who participate in sport, respectively 13.19 vs. 10.87 for TAS F2 and 46.06 vs. 41.87 for the total TAS 20 value.

Several different differences emerged about motivation to participate in sport. Apart from the motivation to control or reduce weight (*p* = 0.002), and to relieving tension (*p* = 0.031), which was slightly more frequent in females, the following motivations were more frequent in males: the desire to stay healthy and fit *(p* = 0.000); pleasure and enjoyment (*p* = 0.000); the acquisition of muscle mass (*p* = 0.000) and the interest in meeting people (*p* = 0.001).

## 4. Discussion

The literature has reported various prevalence rates for the risk of exercise dependence, which are generally below 10%. A meta-analysis in the general population of US adults estimates the risk of dependence at 3% [47]; some studies have suggested a prevalence ranging from 10% to 80% among various sports men and women, such as gymnasts, athletes, marathon runners, etc. [48,49]. Prevalence rates for exercise dependence have been estimated to be between 3% and 9% among athletes, student-athletes and fitness enthusiasts [49]; the prevalence of exercise dependence symptoms (at risk of exercise dependence) in university students was 6% [50]; in a recent systematic review, the prevalence of EDS risk was estimated to be between 3% and 7% of the regular exercisers and university student population, and between 6 and 9% of the athlete population [51]. The percentages for exercise dependence are lower than those for exercise addiction, in fact the Exercise Dependence Scale (EDS) seems to screen a lower percentage of individuals at risk of exercise dependence than the Exercise Addiction Inventory (EAI) [52]. In this research, it is found not only a lower prevalence than exercise addiction, but lower than the cited studies on exercise dependence. In part this could be due to the socio-demographic and cultural characteristics of the sample, which consisted of young university students, for whom sport may not be an exclusive interest. It is also believed that not considering gender and the composition of the sample according to this parameter leads to different percentages. Considering only the male component, our sample is in line with other studies with a prevalence of 3% of subjects at risk for dependence and 47.9% of non-dependent symptomatic.

There are completely conflicting results on the prevalence of dependence between men and women. Costa et al. [53] found that men and young adults had higher exercise dependence scores. Furthermore, for MacLaren, and Best [54], exercise dependence was more frequent in men as well as in all hedonistic behaviours (hedonistic categories include behaviours such as prescription drug use, gambling, caffeine, illicit drugs, alcohol, tobacco and compulsive sex); for Grandi et al. [55] Re-che-Garcia, and Martinez-Rodriguez [50] women had higher scores than men. In our research, there is a male prevalence, but we must point out that in our sample, sports practice is also higher in males. It is believed that this finding is significant, not only to correctly evaluate the prevalence, but also as cultural information.

With regard to sport, our data confirm the advantages of practicing sport, with an increase in levels of subjective happiness, satisfaction with life, and restful sleep. It is also reported a greater ability to identify emotions, but also a greater propensity for externally oriented thinking. It seems to us that these data may be understandable considering that sport can increase relational opportunities, probably stimulating the ability to understand emotions. On the other hand, there is a concrete, operational dimension in sport, which can be found in externally oriented thinking.

A relevant issue in the field of dependence is the assessment of comorbidity, both with respect to dependences and psychopathological symptoms. Among behavioral dependences, substance abuse and gambling are those in which the psychopathological dimension is most relevant [56]. However, comorbidity has also been studied in relation to exercise dependence. The literature has shown significant correlations especially with eating disorders [19,57,58,59]; but other correlations have been found with caffeine use, compulsive buying [54,60], anxiety, hypochondria, time on the computer [28], dysregulation and compulsivity [25]. Exercise dependence would also be accompanied by mood alterations, sleep problems, anxious tendencies and reduced adherence to a low-fat diet [50]; and would have negative effects on self-satisfaction, social behavior and vigor [61]; it would also correlate with psychiatric disorders (depressive, personality, obsessive-compulsive Cluster C [22]. Rather than the risk of exercise dependence itself, we might say that what is of concern is that it may be an indicator of more serious conditions of suffering. However, we should point out that in the literature the correlations were generally weak to moderate (0.06 to 0.28) [34,56] and thus a large proportion of the sample did not present aspects of comorbidity or pathology. Moreover, in contrast, in the work of Li, Nie & Ren [62] exercise dependence appeared to positively modulate state anxiety, depression and subjective well-being. Exercise-dependent subjects had significant reductions in anger and total mood disturbances after physical activity [63]. Our data are quite reassuring, as they showed no other significant dependences or indicators of relationship disorders, and no differences in levels of happiness and satisfaction. 

It is believed that it is not exercise dependence that creates a state of vulnerability for other distressing conditions, but it is the distressing conditions that can find expression in various disturbed behaviors, including exercise dependence. The subjects in our sample are sufficiently happy and satisfied, so it is not surprising that they do not show significant correlations with other indicators of suffering. It is also interesting to note that among the motivations for sporting activity, subjects at risk of exercise dependence and non-dependent symptomatic subjects recognize the function of relieving tension, but also the pleasure associated with the activity. This suggests that the exercise dependence behavior in our sample does not have the characteristics of impulsivity and compulsion, which are generally attributed to dependences [61]. However, it should be noted that the non-dependent symptomatic subjects differ in their increased, albeit moderate, use of soft drugs and the presence of somatic complaints. It would be interesting to follow the evolution of these individuals. Perhaps an evolution towards risk conditions for dependence could reduce somatic symptoms and drug use, functioning as a ‘cure’. Longitudinal research with large samples would be needed to test the ‘curative effect’ or ‘buffer’ of dependence.

It should be noted that apart from the assessment with SWLS, SHS, TAS 20 and EDS other areas were not assessed with specific tests, but with multiple-choice questions on various topics. Moreover, as with any self-report, the answers may not fully correspond to the data that can be collected from a clinical evaluation. However, within the limits of our investigation, it seems more likely to think that students at risk of exercise dependence (and non-dependent symptomatic) do not necessarily suffer from frank psychopathologies, but perhaps face the difficulties of a developmental pathway with an excessive investment in sport. Especially when assessing young people, it is more useful to evaluate not only the fragilities, but also the resources and quality of life, and to monitor the capacity for evolution over time. In this sense, our sample offers positive indicators, but also some concerns, regarding alexithymia. 

The results on this issue are interesting. Firstly, the gender difference is clear. In men, “externally oriented thinking” (TAS F3) is predominant, while in women, scores on “difficulty identifying and distinguishing feelings and physical sensations” (TAS F1) and “difficulty describing feelings” (TAS F2) are higher. Consistent with the literature, which indicates that in behavioural addiction and dependence there is often affective dysregulation and high levels of alexithymia [25,64,65,66] the values for alexithymia (TAS 20 and all three factors of the scale) increase in progression in non-dependents symptomatic and at risk for exercise dependence. But this is only in the sample of men. 

The more operational dimension of thinking (TAS F3), which is more present in men than in women, could be in tune with the physical activity of sport, and somehow, in the case of dependence forms, favour exercise dependence in men and not in women. It should be noted, however, that ‘externally oriented thinking’ (TAS F3) increases significantly in men only in non-dependents symptomatic and those at risk for exercise dependence and does not differentiate men who practise sport from those who do not. This may suggest that the symptomatic and dependent expression of exercise is not so much in continuity with sporting activity, as a quantitative increase, but there is a kind of discontinuity. This may be relevant not only for assessment but also for prevention. It would therefore seem more appropriate to monitor the ability to process and regulate emotions and the quality of thinking, rather than focusing only on the amount of time devoted to sporting activity. In this regard, it is worth recalling the distinction between harmonious passion and obsessive passion: the former makes it possible to increase the time devoted to physical exercise without impoverishing other life activities [67]. It remains to be understood what causes an increase in alexithymia in men at risk for exercise dependence and non-dependent symptomatic men. The hypothesis of a traumatic component is not supported by our data and no other hypothesis can be proposed in our study. 

Certainly, our sample has a favourable characterisation. The subjects attend university, have friendships, find pleasure in sport, are happy and satisfied with their lives. These are important protective factors. Although these favourable conditions define our sample as particular and therefore the results do not apply to the whole population of young people, the fact remains that risk conditions for exercise dependence or non-dependence symptomatic conditions do not in themselves indicate a serious psychopathological condition. Several factors, relating to resources and quality of life, must therefore be considered. It would be interesting to assess whether exercise dependence in young university students represents a symptomatic solution in a critical period, rather than a stable feature, and which factors determine harmonious development. 

It is also believed that the time variable should be considered not only in the individual reading, but also at a social level. Just as the characterization of drug addiction and in general of the consumption of illegal substances has strongly changed in recent decades, and drug addiction is no longer linked to an idea of social and cultural degradation and isolation [68], so it could be for exercise dependence too. This could dilute the characterization, and correlation with pathological aspects. This hypothesis requires a research work prolonged in time and introduces a new variable in the understanding of a complex phenomenon. This research is limited due to the size and characterization of the sample and since it was not desirable to include other tests in the assessment.

## 5. Conclusions

Our data confirm some of the benefits of practising sport. In particular, those who participate in sport have less difficulty “identifying and distinguishing physical feelings and sensations”, fewer somatic symptoms, better sleep, higher scores on happiness and satisfaction with their lives, but higher scores on “externally oriented thinking” than those who do not. On the Exercise Dependence Scale (EDS), the prevalence of subjects at risk of exercise dependence is 1.5% and of symptomatic non-dependent subjects 31.3%. Considering the three categories of the EDS, differences emerged with respect to externally oriented thinking, but no differences for smoking, sleep, traumatic experiences, eating behaviour, friendships, hobbies, levels of satisfaction and happiness. Non-dependent symptomatic subjects have higher levels of soft drug use and the presence of somatic symptoms. Only among men, the percentage of subjects at risk for dependence rose to 3% and among symptomatic non-dependent subjects to 47.9% and there is a progressive increase in alexithymia values (and the three factors) as the EDS categories progress. Women differed from men in lower scores on exercise dependence, use of alcohol, soft and hard drugs, externally oriented thinking and higher scores on TAS F1 (16.28 vs. 13.92) and TAS F2 (13.92 vs. 12.9). There are also differences in motivations to participate in sport. Motivations to control or reduce weight (*p* = 0.002), and to relieve tension (*p* = 0.031) are more frequent in women, whereas the desire to stay healthy and fit, pleasure and fun, increasing muscle mass, and interest in meeting people (*p* = 0.001) are more frequent in men. 

In summary, our data support the hypothesis that the risk of exercise dependence and a symptomatic non dependence condition can occur at a young age in framework of relative well-being (happiness, satisfaction, relationships) without significant associations with other problematic behaviours. The hypothesis of increased levels of alexithymia is confirmed but limited to the male gender. Gender differences are confirmed in the frequency and motivation to practice sport and in the risk of dependence. It is important that the assessment of risk for exercise dependence is integrated with the assessment of alexithymia and personal and social resources over time. 

It is believed that further studies comparing samples from different socio-cultural contexts, with attention to gender differences, with additional assessment tools (not only self-reports) and hopefully also with a longitudinal perspective, could enrich the understanding and curative interventions, where needed.

## Figures and Tables

**Figure 1 ijerph-19-05288-f001:**
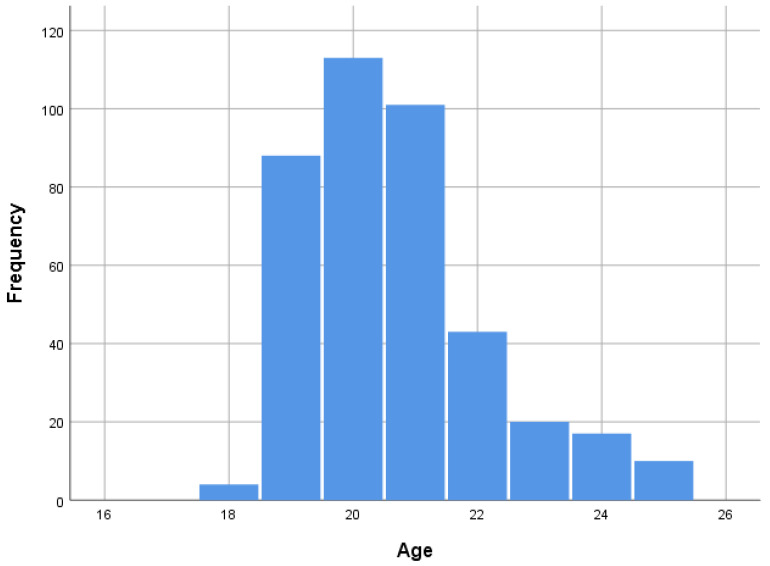
Sample distribution by age.

**Table 1 ijerph-19-05288-t001:** Frequency Exercise Dependence Scale.

Exercise Dependence Scale	N	%
Non-dependent-asymptomatic	258	66.84
Non-dependent-symptomatic	122	31.60
at-risk for exercise dependence	6	1.5
Total	386	97.2

Frequency distributions of the three levels of Exercise Dependence Scale, considering the whole sample.

**Table 2 ijerph-19-05288-t002:** Summary of cases Sport/TAS 20.

Sport	TAS F1 **	TAS F2	TAS F3 **	TAS 20
Sport N. (%)	280 (71.42)	280 (71.42)	280 (71.42)	280 (71.42)
Mean (standard deviation)	14.60 (5.111)	13.44 (4.504)	18.03 (4.455)	45.96 (9.872)
No sport N. (%)	112 (28.57)	112 (28.57)	112 (28.57)	112 (28.57)
Mean (standard deviation)	17.19 (6.551)	13.78 (4.826)	16.40 (4.402)	47.26 (12.255)

Summary of cases. Comparison of those who play and those who do not play sport and TAS 20, in the three factors and in the overall score. TAS F1 difficulty in identifying and distinguishing feelings and physical sensations; TAS F2 difficulty in describing feelings; TAS F3 externally oriented thinking ** *p* = 0.000 Test U Mann-Whitney.

**Table 3 ijerph-19-05288-t003:** Summary of cases EDS/SWLS, SHS, TAS 20.

EDS	SWLS	SHS	TAS F1	TAS F2	TAS F3 **	TAS 20
Non-dependent-asymptomatic (%)	252 (66.31)	256 (69.00)	256 (66.84)	256 (66.84)	256 (66.84)	256 (66.84)
Mean (standard deviation)	23.25 (5.389)	18.34 (4.612)	15.43 (5.904)	13.39 (4.647)	17.15 (4.428)	46.01 (10.909)
Non-dependent-symptomatic (%)	122 (32.10)	119 (32.07)	121 (31.59)	121 (66.84)	121 (66.84)	121 (66.84)
Mean (standard deviation)	22.50 (7.00)	18.45 (4.472)	14.98 (5.013)	13.84 (4.593)	18.45 (4.423)	46.96 (9.929)
at-risk for exercise dependence (%)	6 (1.57)	6 (1.61)	6 (1.57)	6 (1.57)	6 (1.57)	6 (1.57)
Mean (standard deviation)	22.50 (7.00)	16.50 (7.71)	18.00 (8.462)	14.33 (4.412)	22.00 (4.000)	54.33 (11.466)

Summary of cases. Comparison of EDS, and values of subjective happiness (SHS) and satisfaction with life (SWLS). No statistically significant difference; Comparison of EDS and TAS categories in the three factors and in the overall score TAS F1 difficulty in identifying and distinguishing feelings and physical sensations; TAS F2 difficulty in describing feelings; TAS F3 externally oriented thinking ** *p* = 0.002 Test of Kruskal-Wallis.

**Table 4 ijerph-19-05288-t004:** Exercise dependence scale/Soft drug use.

Soft Drug	Never	In the Past Frequently	Few Times	Occasionally	Quite Often	Usually	Tot
Non-depenent Asymptomatic (%)	208	7	30	9	2	2	258
	(80.62)	(2.71)	(11.62)	(3.48)	(0.77))	(0.77)	
Non-dependent Symtopmatic (%)	86	5	12	13	2	4	122
	(70.49)	(4.09)	(9.83)	(10.65)	(1.66)	(1.81)	
At- risk for depedende (%)	3	1	2	0	0	0	6
	(50.00)	(16.66)	(33)	(0)	(0)	(0)	
Total	297	13	44	22	4	6	386

**Table 5 ijerph-19-05288-t005:** Summary of cases EDS/Relieving tension.

	Relieving Tension	No	Total
Non-dependent-asymptomatic (%)	107 (53.23)	94 (46.77)	201
Non-dependent symptomatic (%)	80 (65.57)	42 (34.42)	122
at-risk for exercise dependence (%)	5 (83.33)	1 (16.66)	6
Total	192	137	329

Summary of cases. Comparison of EDS and relieving tension as motivation for physical activity.

**Table 6 ijerph-19-05288-t006:** Summary of cases EDS/Pleasure.

	Pleasure	No	Total
Non-dependent-asymptomatic (%)	120 (59.70)	81 (40.29)	201
Non-dependent symptomatic (%)	104 (85.24)	18 (14.75)	122
at-risk for exercise dependence (%)	6 (100)	0 (0)	6
Total	230	99	329

Summary of cases. Comparison of EDS and pleasure as motivation for physical activity.

**Table 7 ijerph-19-05288-t007:** Summary of cases EDS/ TAS 20.

EDS	TAS F1	TAS F2	TAS F3	TAS 20
Non-dependent-asymptomatic (%)	78 (47.85)	78 (47.85)	78 (47.85)	78 (47.85)
Mean (standard deviation)	12.92 (5.474)	12.17 (4.308)	18.47 (4.351)	43.69 (10.196)
Non-dependent-symptomatic N	80 (49.08)	80 (49.08)	80 (49.08)	80 (49.08)
Mean (standard deviation)	14.54 (4.857)	13.56 (4.443)	19.23 (4.469)	46.83 (9.586)
at-risk for exercise dependence	5 (3.07)	5 (3.07)	5 (3.07)	5 (3.07)
Mean (standard deviation)	19.40 (8.649)	16.00 (1.871)	23.20 (3.033)	58.60 (5.273)
Total	163	163	163	163

Summary of cases. Comparison of EDS, TAS 20, TAS F1 difficulty in identifying and distinguishing feelings and physical sensations; TAS F2 difficulty in describing feelings; TAS F3 externally oriented thinking (men’s sample).

## Data Availability

Data will be available upon reasonable request. Data will be deposited in the university repository, where reasonable requests for access can be submitted.

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
