# Peer review of "Benefits, Risks and Gender Differences in Sport, and Exercise Dependence: Key Role of Alexithymia"

_ijerph, 2022, doi:10.3390/ijerph19095288_

Round 1
Reviewer 1 Report
Starting by congratulating the author for the interesting theme and the quality of the work, I would like to take the opportunity to propose the following improvements:
- review the quality of English, which has some mistakes.
- try to reduce the number of abbreviations, which make the study very difficult to understand.
- lines 33 to 35 - we think it would be important to refer to family life.
- line 121 - "...the Tas 20 [33]. The TAS 20 has good...", needs to be rewritten
- line 170 - It was important to justify the reason for starting only at 18 years old, when we know that before that many young people go to gyms.
- line 172 - "...women. 51.4% were from...", the paragraph cannot start with a percentage.
- line 382 to 385 - this statement must be justified.
- Line 388- "I believe..." should not be used, but the passive form.
- line 382 to 385 - this statement must be justified.
Author Response
I thank the reviewer for his helpful suggestions and his appreciation. I hope that the changes made will adequately address all the comments. Thank you very much.
- review the quality of English, which has some mistakes.- I have checked the English and corrected the errors
- try to reduce the number of abbreviations, which make the study very difficult to understand. - I have entered the full names instead of the acronyms.
- lines 33 to 35 - we think it would be important to refer to family life. - I have included a bibliographical reference on family life
- line 121 - "...the Tas 20 [33]. The TAS 20 has good...", needs to be rewritten. - I have reported the whole quote.
- line 170 - It was important to justify the reason for starting only at 18 years old, when we know that before that many young people go to gyms. – I explained the reasons for the choice of age.
- line 172 - "...women. 51.4% were from..." - The paragraph’s start has been changed
- Line 388- "I believe..." - I changed in the passive form.
- line 382 to 385 - This statement has been justified.
Reviewer 2 Report
The aim of this paper, according to the authors, is to investigate whether high vs. more moderate sport activity levels is part of frameworks of relative well-being rather than in frameworks of suffering or discomfort and if there are differences related to gender. One conclusion is that the risk of exercise addiction can occur in situations of relative well-being (happiness, satisfaction, relationships) without significant associations with other addictive behaviours. The topic is very interesting and the paper is well written, but there are some aspects that should be improved.
General comments
Sport activity levels have not been measured, so the subjects cannot be classified as high or more moderate sport activity. In addition, the questionnaire used is not exercise addiction index, is exercise dependence scale, terms that are sometimes confused but some authors see that they have different nuances. Therefore, authors should consider changing the aim of the study in abstract/introduction, modify title, and adapt the discussion if necessary. In addition, in the text, instead of talking about the risk of exercise addiction, the authors should talk about the risk of exercise dependence.
It is necessary to select mainly references related to exercise dependence and not to exercise addiction.
Abstract
At the end of the abstract, a conclusion related to the main objective of the research should be added.
Introduction
Line 68. The perspective of the study should be changed, since it does not evaluate PA levels or the type of sport practiced by the subjects, so the objective must be modified at the end of introduction.
At the end of the introduction, it is necessary to formulate a statement of the main goal of the research.
Materials and Methods
Line 97. Consider add the full name of the questionnaire before the acronyms. In the text we can see EDS 21 and EDS-21, unify criteria.
Line 121. Consider add the full name before the acronym (TAS).
Lines 128-130. Consider clarifying the acronyms F1, F2 (...) that are used later in the results section and in the tables.
Line 131. “The subjective happiness scale was used to measure the overall level of perceived happiness.” Consider add the acronym in this sentence.
Lines 151-157. It is necessary to briefly explain the questions or questionnaires designed to collect information on: consumption of alcohol, soft and hard drugs, quality of sleep, eating behavior, emotional and friendship relationships, hobbies and the presence of mental or physical disorders.
Results
Consider simplifying and unifying tables. The tables could be simplified if the median data were not added. Mean and the standard deviation could be placed on the same line. Thus, authors will be able to combine tables (such as 2 and 3).
Next to the absolute values, the percentages should be placed to facilitate the reader's understanding.
Line 170. What does the value 68 refer to?
Line 176. How it was determined that a subject was considered in play sports group? Add in the methodological section.
Line 190, 191, 204, 226… How to write p-value? Please unify criteria.
Line 179, 203, 228… How to write “TABLE”? Please unify criteria.
Discussion
Line 268. “The literature reported various prevalence rates of risk of exercise addiction.“ The authors have used the EDS questionnaire and not the EAI (Exercise Addition Inventory). Therefore, data on the prevalence of exercise dependence should be compared with studies that have used the same questionnaire.
Lines 294-326. Summarize this paragraph, highlighting the discussion of the results of this study.
Lines 339-354. It would be better not to talk about factors, rather what they indicate.
Author Response
I thank the reviewer for his accurate remarks, and for his consideration.
In particular, I appreciated the invitation to distinguish between dependence and addiction. I have included a clarification of the terms. I reviewed the literature, added articles and, after re-reading the cited articles on addiction, I eliminated those that really used the EAI.
I added information on the assessment of consumption of alcohol, soft and hard drugs, quality of sleep, eating behavior, emotional and friendship relationships, hobbies, and the presence of mental or physical disorders.
I hope that the changes I have made adequately address all the comments.
Thank you very much.
Abstract
At the end of the abstract, a conclusion related to the main objective of the research should be added.- The conclusions have been added.
Introduction
Line 68. The perspective of the study should be changed since it does not evaluate PA levels or the type of sport practiced by the subjects, so the objective must be modified at the end of the introduction. - As required, I have changed the objective at the end of the introduction.
At the end of the introduction, it is necessary to formulate a statement of the main goal of the research.- The statement of the main goal of the research has been reviewed.
Materials and Methods
Line 97. Consider add the full name of the questionnaire before the acronyms. In the text we can see EDS 21 and EDS-21, unify criteria. -I have added the full name, and used only EDS without “–“.
Line 121. Consider add the full name before the acronym (TAS). - I have added the full name.
Lines 128-130. Consider clarifying the acronyms F1, F2 (...) that are used later in the results section and in the tables. - I have added the full name under the table.
Line 131. “The subjective happiness scale was used to measure the overall level of perceived happiness.” Consider add the acronym in this sentence. – The acronym has been added.
Lines 151-157. It is necessary to briefly explain the questions or questionnaires designed to collect information on: consumption of alcohol, soft and hard drugs, quality of sleep, eating behavior, emotional and friendship relationships, hobbies and the presence of mental or physical disorders. - I have added information about the questionnaire and the response options.
Results
Consider simplifying and unifying tables. The tables could be simplified if the median data were not added. Mean and the standard deviation could be placed on the same line. Thus, authors will be able to combine tables (such as 2 and 3). - Next to the absolute values, the percentages should be placed to facilitate the reader's understanding.- I have modified the tables according to your suggestions.
Line 170. What does the value 68 refer to? - It was the decimals. I changed the spaces.
Line 176. How it was determined that a subject was considered in play sports group? Add in the methodological section.- I have added information in the methodological section.
Line 190, 191, 204, 226… How to write p-value? Please unify criteria. – Criteria have been unified.
Line 179, 203, 228… How to write “TABLE”? Please unify criteria. - Criteria have been unified.
Discussion
Line 268. “The literature reported various prevalence rates of risk of exercise addiction.“ The authors have used the EDS questionnaire and not the EAI (Exercise Addition Inventory). Therefore, data on the prevalence of exercise dependence should be compared with studies that have used the same questionnaire. – I have compared the prevalence of exercise dependence.
Lines 294-326. Summarize this paragraph, highlighting the discussion of the results of this study. – I have modified the discussion.
Lines 339-354. It would be better not to talk about factors, rather what they indicate. - I have added the full name.